# Varnish Formation and Removal in Lubrication Systems: A Review

**DOI:** 10.3390/ma16103737

**Published:** 2023-05-15

**Authors:** Sung-Ho Hong, Eun Kyung Jang

**Affiliations:** 1Department of Mechanical System Engineering, Dongguk University-WISE Campus, Gyeongju-si 38066, Republic of Korea; 2Department of Energy and Electrical Engineering, Dongguk University-WISE Campus, Gyeongju-si 38066, Republic of Korea

**Keywords:** varnish, lubricant, contamination, degradation, oxidation

## Abstract

This study presents the current literature regarding the investigation of varnish contamination among the various types of lubricant contaminations. As the duration of use of lubricants increases, the lubricant deteriorates and may become contaminated. Varnish has been known to cause filter plugging, sticking of the hydraulic valves and fuel injection pumps, flow obstruction, clearance reduction, poor heating and cooling performance, and increased friction and wear in various lubrication systems. These problems may also result in mechanical system failures, performance degradation, and increased maintenance and repair costs. To improve the problems caused by varnish contamination, an adequate understanding of varnish is required. Therefore, in this review, the definitions and characteristics, generating machinery, generating mechanisms, causes, measurement methods, and prevention or removal methods of varnish are summarized. Most of the data presented herein are reports from manufacturers related to lubricants and machine maintenance that are included in published works. We expect that this summary will be helpful to those who are engaged in reducing or preventing varnish-related problems.

## 1. Introduction

Lubricants are widely used in industries such as the automobile, aerospace, and construction industries in addition to power plants. Lubrication is a means to control friction and wear by introducing a friction-reducing layer between moving surfaces that are in contact. Lubricants are typically in liquid or semi-solid form, but they may exist in different forms as solids and gases. This study mainly considered liquid lubricants that perform the following basic functions: lubrication, cooling (heat transfer), sealing, cleaning, damping, and protection from oxidation and corrosion. About 90% of the commercially available lubricants are composed of hydrocarbons, while the remainder includes additives governing their behaviors. The mechanism of lubrication is derived from the physical and chemical interactions between the lubricant molecules, material surfaces, and the environment [1]. Lubrication is often achieved through the physical and chemical properties of the lubricating fluid. Physical properties such as density, viscosity, heat capacity, thermal conductivity, and temperature–pressure–viscosity relationships determine the operating ability of lubricants under hydrodynamic conditions. Chemical properties such as solvency, dispersion, detergency, anti-wear, anticorrosion, frictional properties, and antioxidant capacity are important for successful boundary lubrication. Some of these properties are controlled by the chemical compositions of the base oils whereas other properties are controlled by specifically designed chemical additives [2]. Additives used in lubricants can be grouped into different categories such as friction and wear modifiers, antioxidants, rust inhibitors, anti-form agents, extreme pressure/load-carrying compounds, viscosity index improvers, detergents, dispersants, emulsifiers, and metal deactivators [3,4,5,6,7,8,9,10,11].

As the duration of usage of a lubricating oil increases, it may deteriorate or become contaminated. Due to oxidation and thermal stresses in the working conditions, the physical and chemical properties of lubricating oils may degrade and eventually cause increased wear, which manifest as the depletion of the additives followed by oxidation of the base oil [12,13,14]. The most critical type of deterioration is oxidation, which produces sludge, resins, acids, and carbonaceous deposits because of chemical reactions between the unstable components of the oils and oxygen. Furthermore, the oxidation rate increases in the presence of metallic contaminants and water molecules [15]. Contamination of the lubricating oil also causes deterioration; lubricant contamination may be of internal or external origin. Various contaminants such as dust, water, and humidity are present outside machines and can contaminate the lubricating oils. The contaminants of internal origin are inherent to the functions of the lubricating fluid within the machine during operation and cannot be avoided. These include degradation products or wear particles produced by the machine itself [16]. In addition to the wear particles, the presence of solid particles in the form of dust and rust, which are insoluble in the lubricants, is inevitable. Such undesirable solid particles damage surfaces and may cause fatigue and wear of the machine elements [17]. In addition to solid particles, lubricants may be affected by various contaminants such as moisture, heat, air, antifreeze, solvents, and the improper incorporation of lubricants.

In recent years, varnish-related problems have been increasingly noted in turbine oil during power generation [18,19]. Several names such as sludge, lacquer, goo, gunk, and oil slime have been used to express the deposits found inside lubricated machine systems. Varnish is the most commonly used name for oil-derived deposits. Figure 1 shows examples of sludge on a reservoir and lacquer on a cylinder liner. Sludge is not deposited on metallic surfaces but is suspended in the oil bulk as semisolid black lumps [20]. Sludge can be differentiated from varnish as it is a surface deposit that is easily removed compared to a more tenacious film like varnish. Lacquer appears similar to varnish but is different in that it is difficult to physically separate from the surface due to its strong bonding to metal surfaces; it is also insoluble in most solvents and often removed with acids because it mainly contains quinones [21]. Varnish can cause various problems or damage such as increased wear and filter plugging. The possibility of the occurrence of varnish is usually expressed as the “varnish potential”.

Various cases of varnish contamination have been reported in mechanical systems or components. Approximately 40% of the 192 power plants surveyed by Exxon Mobil were reported to experience varnish-related problems [22]. Gas turbines (GTs) are particularly sensitive to varnish-induced valve sticking, which causes unit trips and fail-to-start conditions. The production loss caused by a single event can easily cost a typical GT operator up to $480,000 a day [23]. In some turbines, varnish causes the servo valves to stick or may even cause a complete shutdown. The associated downtime and repair costs may vary from $100,000 to millions [24]. GE (General Electric) reported that gas turbines showed signs of oil varnishing from about a decade ago and that their OEM (original equipment manufacturer) recommended using varnish removal systems [25]. Varnish problems have also been reported in bearings [26,27], gears [27], gaskets and seals [28], storage tanks and oil flow lines [29], filters [30], and piston rings [31]. Hence, varnish-related problems may occur in both the mechanical elements of turbines and other mechanical systems, incurring high maintenance and repair costs. Therefore, we review current studies and reports on varnish-related problems to present the definitions, characteristics, occurrence mechanisms, causes, measurement methods, and removal methods of varnish while providing useful information to researchers and concerned entities in developing appropriate solutions. 

Furthermore, it is intended to present a new direction for research on varnish such as evaluating varnish contamination and solving problems caused by varnish at an early state by using condition diagnosis technology based on oil sensors, which is widely applied today.

## 2. Varnish

### 2.1. Definition and Characteristics

Varnish is a thin, hard, lustrous, and oil-insoluble deposit composed primarily of organic residues. Moreover, varnish is a soft contaminant composed of the byproducts of lubricant degradation of less than 1 micron in size that cannot be measured by traditional particle counters [25,32]. Varnish can also be considered as both a soluble and insoluble contaminant comprising the byproducts of oil degradation [31]. As previously mentioned, varnish is different from sludge or lacquer and should not be used with the same meaning. The quickest method of identifying varnish is by its color intensity; varnish can be observed to have various colors depending on the viewing angle. It is most often light-orange, brown, or black in color. Lacquers of a similar type do not dissolve in solvents but varnish can be easily removed by wiping with a solvent such as acetone or ether.

Varnish is a sticky deposit that adheres to the metal surfaces, so that solid particles like dirt and wear particles are trapped in the deposit. These hard particles then cause abrasive wear and act as catalysts for oil oxidation [31]. Studies investigating varnish have traditionally focused on insoluble deposits evidenced by the problems faced, but varnish is initially an agent of oil-soluble degradation. In other words, varnish has different forms and is either insoluble (particulate) or soluble (dissolved) in lubricants [32]. Hence, the solubility of varnish precursors in lubricants is important. The solvency of lubricating oil is mainly affected by its molecular polarity, contaminant level, and temperature [25]. Varnish precursors formed by the oxidative degradation of oils have finite solubility in a lubricant’s nonpolar mineral-oil matrix. Degradation products that are more polar are correspondingly less soluble. The most basic concept of solvency can be stated as “like dissolves like”.

Metals are more polar than the base stocks of lubricants, so the polar varnish precipitates adhere to the metal to form deposits. Lubricating oils have a finite capacity to dissolve other particles such as additives, contaminants, and varnish precursors. When the oil degrades and oxidation products accumulate, the fluid solvency decreases accordingly. Beyond the saturation point, the fluid no longer dissolves additional varnish precursors, and the varnish precipitates as a solid. Temperature also affects the solvency of the lubricants. As the temperature decreases, the solubilities of varnish and its precursors also decrease. Moreover, varnish is naturally polar and can be removed from the system by dielectrophoresis [32,33]. Dielectrophoresis is a phenomenon in which dielectric particles in a nonuniform electric field experience a force; this force even acts on uncharged particles.

All particles exhibit dielectrophoretic activities in the presence of an electric field. However, the strength of the force is strongly dependent on the medium (lubricants), the particles’ electrical properties, shapes, and sizes as well as the frequency of the electric field [34]. The polar nature of varnish thus helps its removal and promotes oxidation by attaching the wear particles to its surface.

### 2.2. Varnish-Generating Machinery

Varnish is generated by various mechanical components, as shown in Figure 2. The mechanical system with the greatest varnish formation is the turbine. Furthermore, varnish-induced machine failures have become critical in recent years given the long-life turbine oils used in thermal power generation. To achieve higher operation efficiencies, the equipment designs are being changed in turbines. Turbines operate at higher temperatures, so their lubricants must also operate at high temperatures. Moreover, thermal problems may occur because of the smaller footprint and lower-volume lubricant reservoirs. Varnish might also occur due to the combination of the turbine’s bearing and control oils. Operating and equipment design changes place additional oxidative and thermal stresses on the turbine oil, which can result in premature aging and degradation [35]. Varnish also causes problems such as sticking and the faulty operation of the control valves, higher bearing temperatures, bearing failures, oil filter blockage, and poor heat transfer [29,36,37]. In addition, varnish can plug the oil inlets and strainers as well as orifices while restricting oil flow in pipes, damaged mechanical seals, and gear and shaft formations [19,38].

### 2.3. Generation Mechanism

Varnish can exist in both soluble and insoluble forms. Lubricants have finite abilities to dissolve varnish at given temperatures. When soluble varnish levels reach the solvency capacity, the solutions are saturated. At the saturation point, a lubricating oil cannot dissolve additional varnish. The equilibrium point between the relative levels of soluble and insoluble varnish is affected by the temperature, molecular polarity, and contamination level [25]. Figure 3 shows the process of varnish generation. During operation, lubricants are chemically deteriorated by oxidation, heat, and micro-dieseling. This process (Step 1) is irreversible, producing soluble varnish that accumulates in the solution. As the soluble varnish accumulates, the lubricant’s saturation point is eventually achieved, beyond which any additional varnish produced is insoluble because the lubricant’s capacity to hold varnish is exceeded. Therefore, the further degradation of saturated lubricants produces insoluble varnish particles (Step 2); then, the insoluble varnish particles eventually agglomerate into deposits (Step 3). The phase changes between the insoluble and soluble varnish are physical and reversible. In addition, most of the lubricant breakdown occurs in the hottest parts of a system.

These high-temperature regions are sufficient for lubricant heating and its increased capacity for dissolving soluble varnish. When the varnish subsequently cools in the other areas, the oil saturation point decreases. Although lubricating oils accumulate soluble varnish when warm, the concentrations often exceed the lubricant capacities in the cooler areas. When the soluble varnish level in lubricants exceeds its saturation point, the solvent is said to be supersaturated. The soluble varnish is converted to insoluble varnish and deposited until its level reduces below the manageable amount at a particular temperature. In the absence of methods to reduce the specified soluble varnish levels at given lubricant operating temperatures, the varnish continues to be precipitated and deposited in the cooler areas [23,29,39].

### 2.4. Causes

Figure 4 shows the causes by which varnish occurs; varnish is primarily a result of oil degradation caused by oxidation. Oxidation is a chemical reaction between dissolved oxygen and the base oil at high temperatures. During oxidation, hydrocarbons break down, forming reaction products called radicals. The subsequent reactions of these radicals form peroxides and must hence be quenched by antioxidants to preserve the lubricant’s integrity or its remaining useful life (RUL). The role of an antioxidant is to protect the base oil by either scavenging radicals or decomposing hydroperoxides into stable products. The depletion of antioxidant additives can produce insoluble varnish. Oxidation accelerators such as heat, water, and metals are important as they can act individually or in combination, but they are application dependent [40].

The elevated temperatures of oils and lubricating surfaces can accelerate oil oxidation. This may be attributed to local hotspots such as local bearing overheating or overall high operating temperatures. According to Arrhenius’s law, the lubricant lifecycle is halved for every 10 °C increase in temperature, meaning that the oxidation rate doubles for every 10 °C increase in operating temperature. This rule is inexact but useful, as it indicates the impact of high temperature on the oil oxidation rate [19].

Moisture/water contamination may occur by ingestion and condensation, acting as an oxidative accelerator. The water volume that dissolves in the oil depends on the base stock, additive package, contamination level, and temperature of the oil. Water exists in three forms, namely, free, emulsified, and dissolved water. Typically, new high-grade oils with minimal additive loads hold little dissolved water; in contrast, oxidized lower-grade oils hold up to 2000 ppm of water in the dissolved state. In this state, the water is not visible in the oil; however, once the water content exceeds the threshold for dissolution, the oil becomes saturated. At this point, water is suspended in the oil in the form of microscopic droplets, which is called an emulsion. Emulsified water often has a hazy appearance; adding more water to an emulsion can cause separation of the two phases, producing a layer of free water. Free water separates from the oil dues to its inherent insolubility and the difference in specific gravity between the two fluids. In most cases, free water is found at the bottom of tanks and sumps [40].

Solid contamination by wear debris or dirt ingestion accelerates oxidation by acting as a catalyst and decomposes hydroperoxides. Iron and copper are good examples of wear metal catalysts in oxidation. Wear debris is generated by numerous wear modes such as abrasive, erosive wear, fatigue, and corrosive wearing. These mechanisms can generate particulate contaminants that further cause component damage. The presence of particulate contamination can cause varnish formation [40,41,42].

Micro-dieseling occurs when small air bubbles trapped in the turbine oil are compressed, causing an explosion and oil burning, as shown in Figure 5a. This often occurs in pumps or bearings and causes adiabatic compression with temperatures of about 1000 °C to create submicron-sized carbonaceous deposits. That is, these temperatures are high enough to carbonize oil at the bubble interfaces, resulting in carbon byproducts and increased oil degradation. Micro-dieseling also creates a dark brown varnish; if the bubble amounts in the oil can be controlled, then micro-dieseling can be controlled [42,43].

Spark discharge generated in the oil filter is one of the causes of hotspots that induce thermal degradation. Studies have observed spark damage and burnt oil inside the oil filters as well as varnish accumulation on the outside. Figure 5b shows microscopic nylon balls created by static discharge; the local oil temperature increases up to 20,000 °C, and a subsequent spark discharge cracks the oil molecules to create free radicals that polymerize into varnish [37,44,45].

Improper synthesis is also a method of varnish formation, which reacts with the lubricating oil to produce precipitates. Even if particulates are not formed, incompatible liquids such as improper oils and solvents can impair the lubricating oil through interactions with the other contaminants. In some cases, changing to a Group II oil without testing the compatibility with a previously used Group I oil can cause decomposition and/or fresh additives to remain from Group I for unexpected reactions with Group II additives. Furthermore, preservative fluids used for corrosion prevention in new equipment may not be compatible with the turbine oil. Turbine oils are typically ashless, and the metals in some preservative and flushing fluids can react with the acidic components of the turbine oils to form insoluble soap, which forms varnish [37,42]. Ultraviolet (UV) degradation is another cause of varnish formation. Although most lubricating oils are not exposed to UV (sunlight) for degradation, UV light can degrade new oils stored outside in oil reservoirs such as polytotes; UV thus causes rapid oil degradation, contributing to varnish formation [42].

There are five classifications of base stock in API (American Petroleum Institute). Group 1 is the least refined and usually a mix of different hydrocarbon chains. These oils are generally used in applications without high performance demands. Group I oils are more polar in nature. This polarity leads to higher solvency than other mineral oil base stocks. Group II and Group III oils are more refined than Group I, and are more oxidatively stable. However, the additives needed to increase solvency deplete, the other additives begin to separate and form varnish. That is, the group of base oil also affects the formation of varnish [38].

### 2.5. Measurement Methods

There are various measurement methods for varnish potential such as oxidation stability, deposit measurement, contamination level, and measurement of the molecules or atoms of the contaminants, as shown in Figure 6. In addition, the form tendency and air release test, water content, and simple deposit tests such as the panel coker and hot micro-coking have been applied to measure the varnish potential of the lubricants.

The membrane patch colorimetry (MPC) test has been adopted as an industry standard to assess an oil’s potential to form harmful varnish [46]. In this test, an in-service lubricant sample is passed through a fine filter patch (0.45 μm), and the color of the remaining organic residue (varnish) is quantified. Darker and more intensely colored residues indicate degraded oils with a high potential for varnishing, as shown in Figure 7. When using MPC testing, it is important to follow the ASTM method to maintain consistency in the test results, which can otherwise vary with the duration of storage of the oil sample. The standard MPC test method requires that all samples be heated to 60 °C for 24 h and subjected to an incubation period of 68–74 h in the absence of light; this is because hydrocarbon-based lubricants continue to break down after the sample is drawn. This process ensures that prior to analysis, all samples are reset to similar starting points. Although these steps necessitate a 96 h waiting period, they are required to obtain reproducible and useful evaluations of the varnish potential of a fluid [23,32,39]. Similar to the MPC approach, the colorimetric patch analyzer (CPA) method can also be used to measure the varnish potential. The CPA uses the membrane patch color using both reflected and transmitted light waves, whereas conventional colorimetric analyzers use only reflected light, as shown in Figure 8a. Figure 8b shows a schematic of the filtering equipment. Figure 8c presents a magnified view of the surface of a membrane filter and its cross-sectional structure. The membrane filter is clamped between the filter support of the vacuum flask and filter funnel, and about 25 mL of the sample oil is filtered through the funnel at reduced pressure. The membrane filter has a pore size of 0.8 μm, a diameter of 25 mm, and a thickness of 0.125 mm. In addition, prior to filtering, the sample oils are heated to 60–65 °C for one day with continuous stirring, followed by incubation and storage at room temperature away from UV light for three days, in accordance with the guidelines in Section 8.1 of ASTM D7843 [36,47]. The CPA has a similar oil sample preprocessing method as the MPC, but the results are evaluated differently as RGB values.

The rotating pressure vessel oxidation test (RPVOT, ASTM D-2272) and RULER (voltammetric techniques, ASTM D-6971) are useful for evaluating the remaining life of the turbine oil based on the remaining lifetime of its antioxidants. The RPVOT is an indirect method for evaluating the residual lives of turbine oils, whereas RULER is a direct antioxidant measurement method. The RPVOT is also known as the rotating bomb oxidation test (RBOT), which involves placing an oil sample in a rotating pressure vessel along with a given concentration of water and a copper coil. The vessel is then pressurized at 90 psi with pure oxygen and placed in a heat bath at 150 °C on a device rotating at 100 rpm. As the temperature of the pressure vessel and its content increases, the pressure increases until it stabilizes, after which the test starts. During the RPVOT, the oil’s ability to resist oxidation degrades by stress-induced antioxidant depletion to the point where the base oil reacts with oxygen as the oil molecules are oxidized. At this point, the pressure drop in the pressure vessel accelerates; when the pressure reaches 25 psi, this is known as the endpoint of the RPVOT. The RPVOT residual rate is the ratio of values of the used to new oils, which is expressed as a percentage, as shown in Equation (1); RPVOT(t) is the value after t hours of degradation [40,48,49]:(1)$$\mathrm{RPVOT}\;\mathrm{residual}\;\mathrm{rate}=\frac{\mathrm{RPVOT}\;\left(t\right)}{\mathrm{RPVOT}\;\left(\mathrm{new}\right)}\times 100$$

However, this technique has some limitations: the main among them is the reproducibility of ±22%, meaning that the test is designed to measure a large drop over time rather than very small iterative changes [49].

The RULER is used to assess the degree of antioxidant activity related to the varnish of lubricants. The RULER method is based on voltammetric analysis, in which the oil sample is mixed with an electrolyte and a solvent before being placed in an electrolytic cell to detect the electrochemical (antioxidant) activity. The oil samples are diluted in a mixture of acetone/electrolyte to enhance antioxidant extraction into the solvent phase. During voltammetric analysis, the potential across the electrodes varies linearly with time, and the resulting current is a function of the potential. When applying an increased voltage to the sample in the cell, the various additive species under investigation oxidize electrochemically. The data recorded during this oxidation reaction is then used to predict the RUL of the lubricants or evaluate the remaining antioxidant additives in the used samples [41,50].

The turbine oil stability test (TOST) is used to assess the degree of antioxidant activity for the turbine oil. The test involves exposing the oil to stresses that promote oxidation and deposition using oxygen, water, high temperature, and metallic components. A total of 60 mL of distilled water is added to 300 mL of the oil sample that is to be tested, which is then constantly heated to 95 °C. In addition, steel and copper wire coils are suspended in the test tank as catalysts, with 3 L of oxygen being passed through the oil–water mixture every hour. The test ends once an acid value of 2.0 mg KOH/g is attained. The time elapsed at this point is stated in hours as the test result. The smaller the value of the result, the higher the tendency of the oil to oxidize rapidly. Recently, dry TOST is widely used for the oxidation test of turbine oils and is a modified version of the conventional TOST; this method uses the same apparatus as the conventional ASTM D 943 TOST, but with two important differences. First, water is not used in the test; second, the temperature is set to 120 °C. As above-mentioned, the ASTM D 943 measurements are typically conducted with 60 mL water by maintaining the temperature at 95 °C [50,51,52,53].

Fourier transform infrared (FTIR) spectroscopy is used to measure additive depletion, contaminants, and base-stock degradation in the lubricants. This method is also used to measure the varnish potential. The test principle involves an infrared absorbance spectrum that is acquired by passing infrared light through a thin layer of the static sample. The chemical constituents of the sample absorb some of the infrared light at reproducible and specific wavenumbers. A computer algorithm called fast Fourier transform is then used to convert this signal to an absorbance spectrum. FTIR analysis is a powerful tool for identifying molecular changes in lubricants during degradation. Varnish is closely related to oxidation; as oxidation increases, the typical reaction byproducts obtained are carbon–oxygen double bonds, also known as carbonyl groups. As the oxidation increases, the absorbance peaks increase. Additionally, phenol inhibitors that are used as antioxidants in the oil show changes in their observed peaks [29,54,55,56,57].

Differential scanning calorimetry (DSC) is used to characterize mixtures of lubricants and estimate their varnish potentials. One drop of varnish is weighed and placed in an aluminum crucible. The low viscosity of the sample enables good dispersion in the crucible. The sample is then placed in an oven at 50–60 °C for 10 days to ensure slow evaporation of the solvents without the formation of parasite bubbles. The dried sample is then weighed and analyzed by DSC using temperature programming. DSC measures the transformation energies undergone by a material subjected to temperature variations. The DSC is used in determining the melt percentage of lubricants by measuring the enthalpy changes. For melting lubricants requiring energy E (Equation (2)), the thermodynamic expression is as follows:E = m_smp_ ∆H = m_lub_ × ∆H_flub_
(2)
where m_lub_ is the mass of the lubricant mixture in the varnish sample; E is the total energy in joules associated with the melting of all compounds; m_smp_ is the mass of the varnish; ∆H_flub_ is the melting enthalpy of the mixture of lubricants [58]. 

The acid number (AN), previously referred to as the total acid number (TAN), is used to estimate the varnish potential. The AN is a measure of the amount of acidic substances in the oil and is an indicator of lubricant degradation. The AN increases with time due to oxidation. AN monitoring is therefore used as a measure of the risk of oxidation and varnish [59,60].

The ultracentrifuge (UC) test measures the concentration of oxidation byproducts and insoluble varnish in the oil. A small amount of oil is placed in a test tube and processed for 30 min at 18,000 rpm in an ultracentrifuge. By subjecting the sample to a high centrifugal force, the oil degrades the insoluble contaminants that are too small to be detected by normal particle counters or removed by mechanical filters. Tracking the ultracentrifuge ratings is a measure of the tendency of the oil to form oxidation byproducts, leading to varnish and sludge. The density of the agglomerated material is compared using a rating scale to obtain the UC value ranging from 1 to 8, where 1 is the best rating and 8 is the worst rating. The lube oils are drained and the test deposit compared against known references, as shown in Figure 9 [61,62]. 

Furthermore, particle counting methods such as the NAS 1638, ISO 4406, and SAE AS 4059 are utilized. The Karl Fischer method for measuring the amounts of water, foam tendency, and air release; inductively coupled plasma (ICP) for elemental analysis; viscosity are used to estimate the varnish potential [18,19,29,30,33]. Deposit tests are performed to measure the varnish potential, as shown in Table 1. 

### 2.6. Prevention and Removal Methods 

There are several varnish-related problems in industrial oil systems. Figure 10 shows four approaches that can reduce or mitigate these problems. Maintenance strategies such as preventive or predictive maintenance are used to manage many systems to maintain good reliability. These techniques are related to condition diagnoses such as noise and vibration, thermography, and lubricant analysis monitoring. The damage caused by varnish contamination can be prevented by regularly analyzing the properties of the lubricants and its contamination levels. Both the offline method of analyzing the lubricating oil through oil sampling and online or inline method using sensors are widely used [68,69]. When varnish contamination occurs in hydraulic oil, the dielectric constant shows distinct variations compared to that of the new oil because the dielectric constant is largely expressed as a sum of two types of polarizability and a dipole moment, where the generation of varnish affects the dipole moment due to oxidation [69,70]. That is, the possibility of predicting varnish contamination is given by measuring the dielectric constant through a lubricant sensor. However, additional research is needed to provide a clear criterion for varnish contamination.

Next, lubricating oils have been used to lower the possibility of varnish; the flushing method in the oil replacement process and various other physical, chemical, and electrical methods that are used to reduce and mitigate varnish problems are shown in Figure 10. As the usage period of the lubricating oil increases, its performance deteriorates, so it is necessary to replace the lubricants at appropriate intervals to ensure appropriate functioning and effective management. In the process of replacing the lubricating oil, the process of oil flushing involves the removal of the used oil and cleaning of the remaining foreign substances or sludge [71]. Flushing can remove deposits, which are the precursors of varnish. Moreover, cleaning chemicals containing dispersants and detergents are used during flushing to effectively remove varnish. While cleaners are very effective at removing varnish and deposits from internal surfaces, it is important to flush out all traces of the chemical cleaners before the system is refilled with fresh oil [72]. The reason for this is that any remaining chemical cleaners may react with the additives in the lubricants or change its properties. Varnish formation is related to the solvency of the lubricants, so adding a solvency enhancer to the oil will increase its ability to redissolve varnish and bring it back into the solution.

Solvency improvers are effective but not very aggressive at cleaning varnish from surfaces. They work better as varnish preventatives by maintaining the varnish precursors in solution in the oil and slowing the formation process [73,74]. Table 2 shows examples of chemical cleaners and solvency improvers that are being used to remove and mitigate the occurrence of varnish [75,76]. In addition, the physical, chemical, and electrical methods to reduce and mitigate varnish problems include depth filtration, electrostatic oil cleaning, charged agglomeration cleaning, resin-based chemical absorption, and ion-charge bonding.

Depth filtration

A filter can be placed in the lubrication circulation loop to remove varnish particles. This filter includes surface (membrane) and depth filters, as shown in Figure 11. While the surface filter retains the particles on the surface of the media, the depth filter retains the particles either in a thicker medium or in multiple layers of media. Furthermore, the depth filter is capable of handling a larger amount of particulates of different sizes and is more cost-effective than the surface filter; however, it can be difficult to remove smaller particulate matter [77]. Depth filtration only removes the degradation products in the suspension but not in the solution. Hence, depth filtration is not very effective when used alone and must be combined with another removal technology to ensure good performance.

2.Electrostatic oil cleaning

Electrostatic oil cleaning can remove varnish via an electrostatic charge because varnish is naturally polar. A high-voltage no-current electrostatic field is maintained across the electrode. When polar varnish and hard particles pass through an electric field, they are attracted to the negative or positive electrodes, whichever is oppositely charged to the particle’s own charge, as shown in Figure 12. This is similar to a magnet being attracted to the opposite pole of another magnet. This method can dissolve varnish on the surfaces of oil circuits and reestablish equilibrium between the varnish and its precursors. Moreover, this method removes the degradation products in suspension only, so if used in combination with another technique, the efficiency can be increased. Electrostatic oil cleaning is sensitive to water and conductive contaminants because water compromises the electrostatic field by carrying current, and the conductive contaminants do not easily attach to the poles [72,78,79]. 

3.Charged agglomeration cleaning

This method is similar to electrostatic oil cleaning, where the particles are charged with electrostatic forces and then allowed to agglomerate. However, the charge on the particles is not restrained to the confines of filtration. In contrast to electrostatic oil cleaning, agglomeration occurs when the oil returns to the lubrication system. The submicron particles agglomerate into multimicron-sized particles, which can be filtered mechanically. In detail, the process involves dividing the fluid into two streams, and the particles are charged positively or negatively in separate flow streams, as shown in Figure 13. When the charged particles are recombined downstream, they form natural and larger particles that can be removed by conventional mechanical methods such as the depth filter. This method is not only efficient at removing suspended contaminants but also sensitive to water and conductive contaminants [43,72,73]. 

4.Resin-based chemical absorption

This method removes varnish by absorbing the soft contaminants via ion exchange in the resins. That is, the oil passes over a resin bed, and the contaminants are absorbed by ion exchange. The process is also known as electrophysical separation and ion-charge bonding (ICB). It uses billions of sites in the bed that are capable of adsorbing soluble varnish. This adsorption relies on preferential interactions between the varnish molecules and sites containing ICB media. It does not involve filtration but removes varnish by chemically bonding the varnish to the surface of the adsorption medium. Selective ion-exchange resins are mixed and formulated to absorb within their porous structures. Their absorptive nature is attributed to the polar attraction between the ion-exchange resin and varnish contaminant. This prevents soluble varnish from accumulating in the lubricants and eventually forming harmful varnish particles and deposits [23,39,72,78].

Table 3 shows the characteristics of products related to varnish removal such as depth filtration, electrostatic oil cleaning, charged agglomeration cleaning, and resin-based chemical absorption. Although the products corresponding to the technologies shown in Table 3 are not the most representative, their characteristics can be roughly compared. The operating temperature, pressure, treatment capacity, and resolution of the treatable particle sizes vary greatly depending on the removal technology used. That is, an appropriate varnish removal technique is applied according to the characteristics of the system in which the varnish is generated. Moreover, for system efficiency, products that combine several technologies for use in varnish removal systems are also currently available. To reduce or solve varnish problems, it is necessary in the near future to improve the performance of each technology and develop diagnostic methods for varnish contamination in real-time using lubricant sensors.

Although there are various techniques for removing varnish as described above, however, the problems caused by varnish contamination have not yet been well-resolved. One of the reasons is that regular offline analysis method such as MPC are not effective to diagnose varnish contamination immediately. Therefore, there is a need to develop a method for accurately measuring varnish contamination in real-time by using a lubricant sensor. Moreover, it is also necessary to reduce the maintenance costs by improving the durability and performance of the filtering systems.

## 3. Conclusions

As the service life of lubricating oil increases, the oil itself deteriorates and contamination occurs. Among the problems caused by lubricant deterioration, varnish contamination is a problem in systems such as turbines used for power generation. The varnish causes filter blockage, sticking of the fuel injection pumps and hydraulic valves, poor heating and cooling performances, clearance reduction, an increase in friction and wear, and flow obstruction in orifices. This study investigated the previous literature regarding mitigating and improving varnish-related problems in lubrication systems. The cited references include research papers and many technical reports from lubricant manufacturers, power generation companies, and companies producing refining systems for lubricants. This literature review summarizes the mechanical elements by which varnish occurs, their definitions, causes and occurrence mechanisms, and measurement methods for varnish. In addition, it introduces four approaches for mitigating and reducing varnish and explains the related techniques. This review also presents examples of products to which the related technologies are applied and compares their characteristics. To effectively solve varnish-related problems, it is necessary to develop an integrated system in which various technologies are combined to suit the system characteristics as well as develop individual technologies for removing or mitigating varnish. Basically, it is necessary to improve the performance of filtering systems. Moreover, a device that effectively removes varnish in a melted stat as well as a technology for diagnosing varnish contamination at an early stage using lubricant sensors should be combined. Finally, it is also essential to develop effective lubricant sensors and diagnostic algorithms to diagnose varnish contamination in real-time.

## Figures and Tables

**Figure 1 materials-16-03737-f001:**
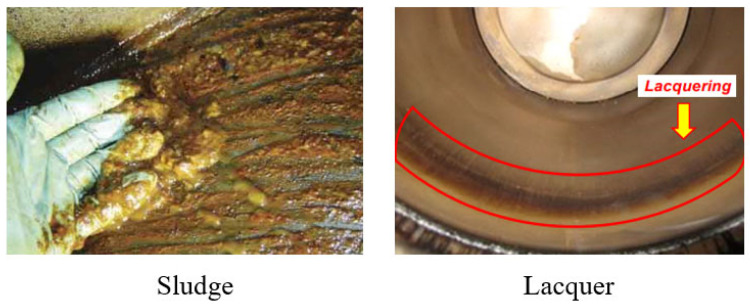
Sludge and lacquer [20,21].

**Figure 2 materials-16-03737-f002:**
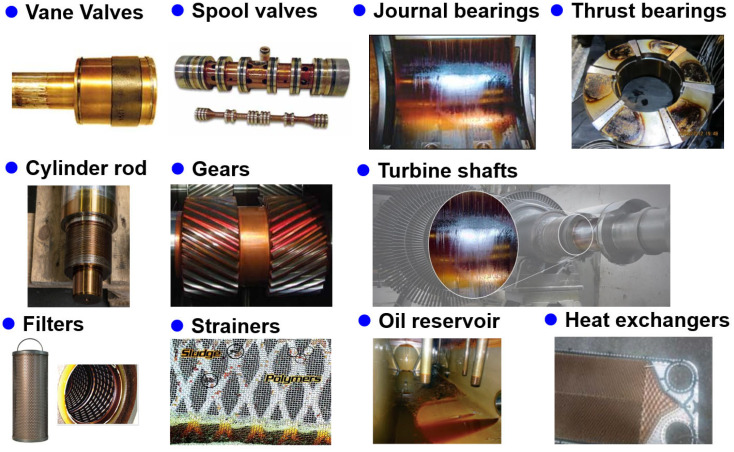
Varnish formation in various components [18,19,26,27,28,29,30,37,38].

**Figure 3 materials-16-03737-f003:**
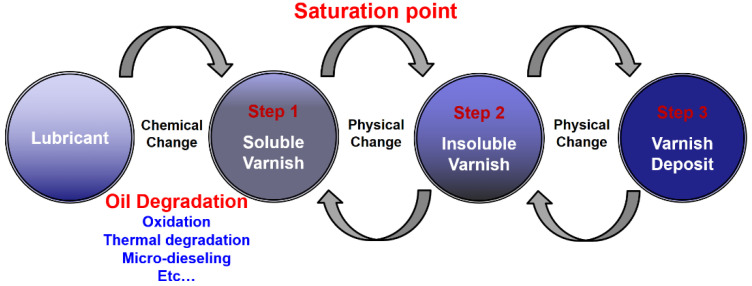
The varnish generation process.

**Figure 4 materials-16-03737-f004:**
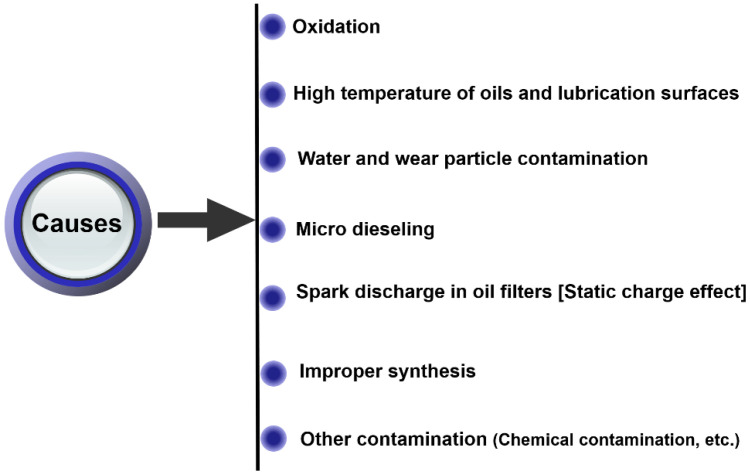
Causes of varnish generation.

**Figure 5 materials-16-03737-f005:**
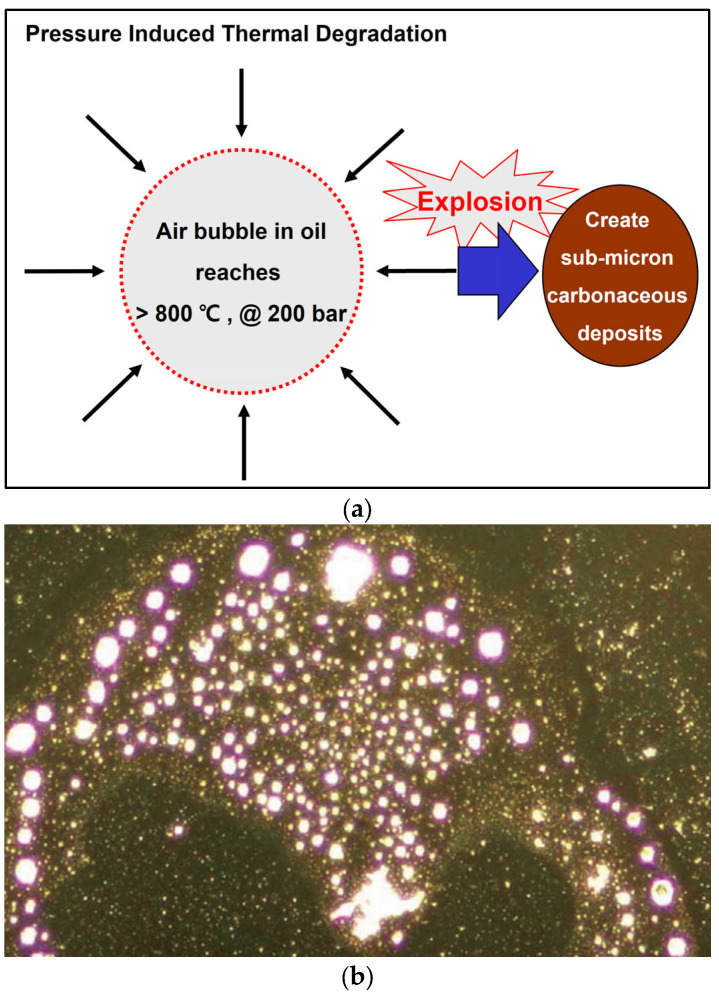
Micro-dieseling and sparking of filters: (**a**) micro-dieseling; (**b**) fine balls of nylon produced by spark discharges [37].

**Figure 6 materials-16-03737-f006:**
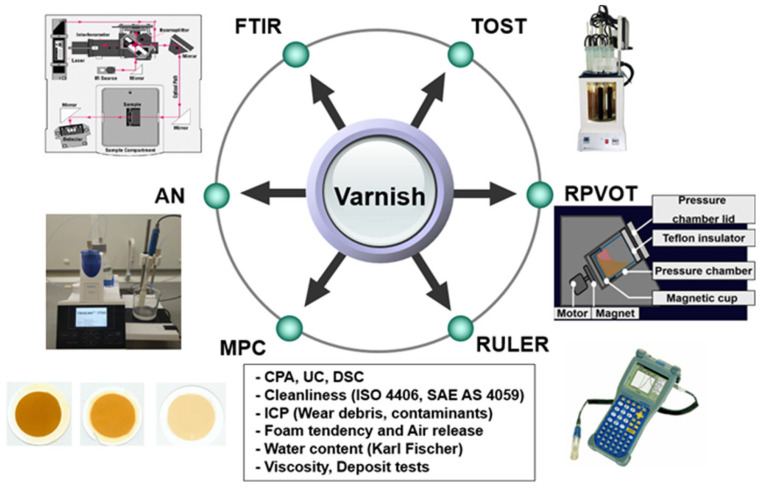
Methods of measuring varnish potential.

**Figure 7 materials-16-03737-f007:**
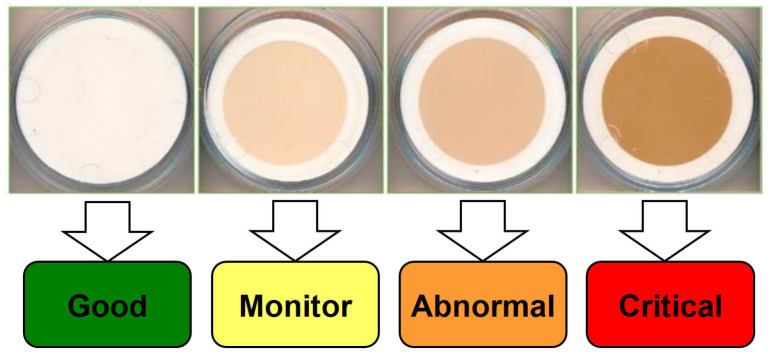
Varnish potential assessment by MPC [36].

**Figure 8 materials-16-03737-f008:**
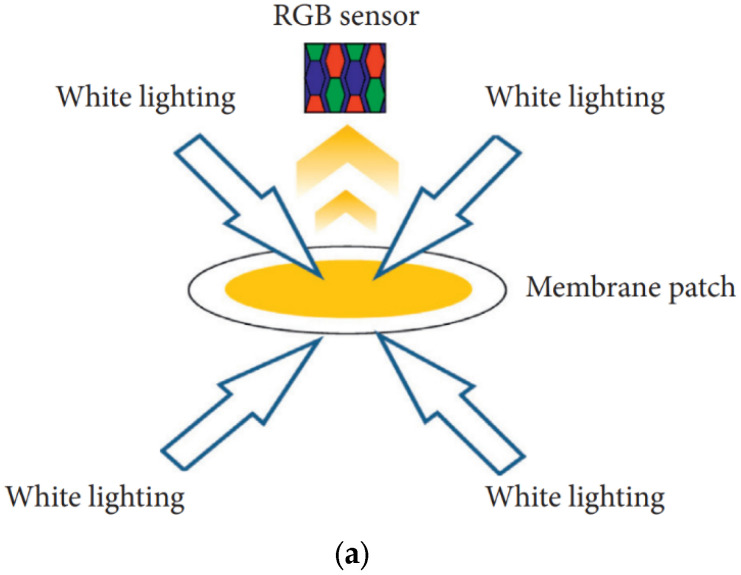
CPA for the measurement of varnish potential: (**a**) measurement principle; (**b**) filtering equipment; (**c**) magnified images of membrane filter [36].

**Figure 9 materials-16-03737-f009:**
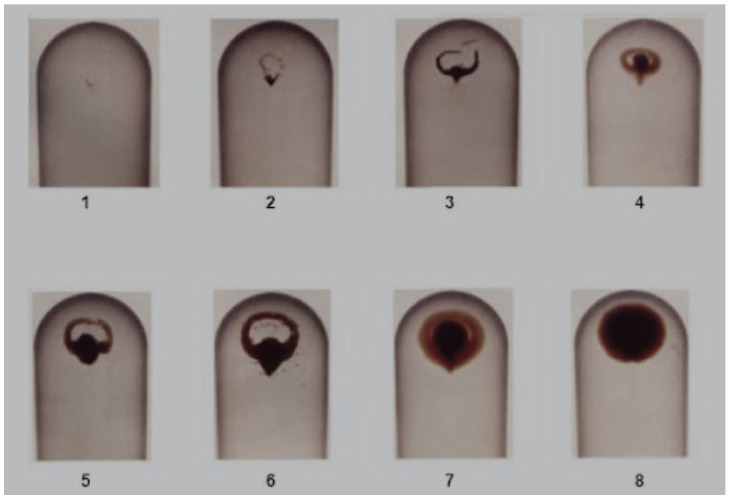
Reference numbers of the ultracentrifuge (UC) test [61].

**Figure 10 materials-16-03737-f010:**
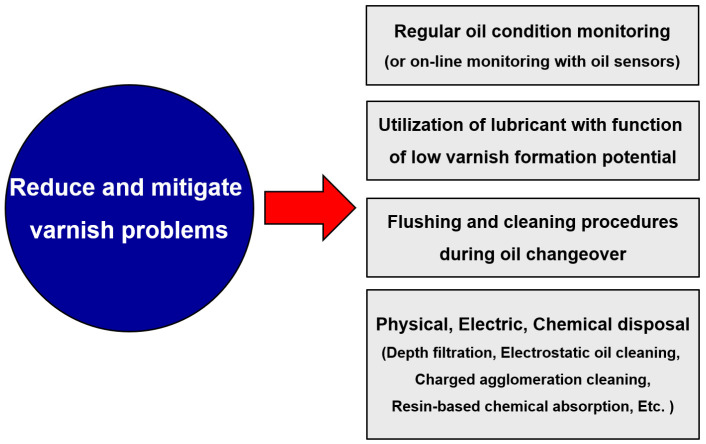
Mitigation strategies for varnish problems.

**Figure 11 materials-16-03737-f011:**
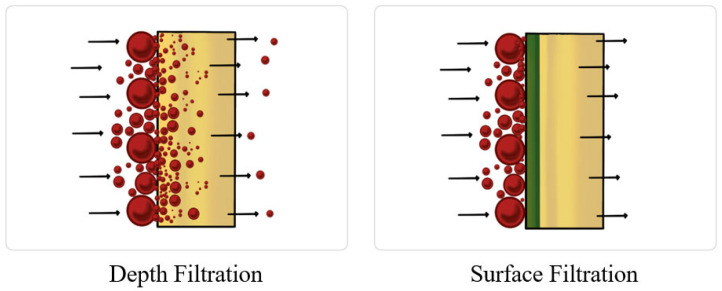
Depth filtration and surface filtration.

**Figure 12 materials-16-03737-f012:**
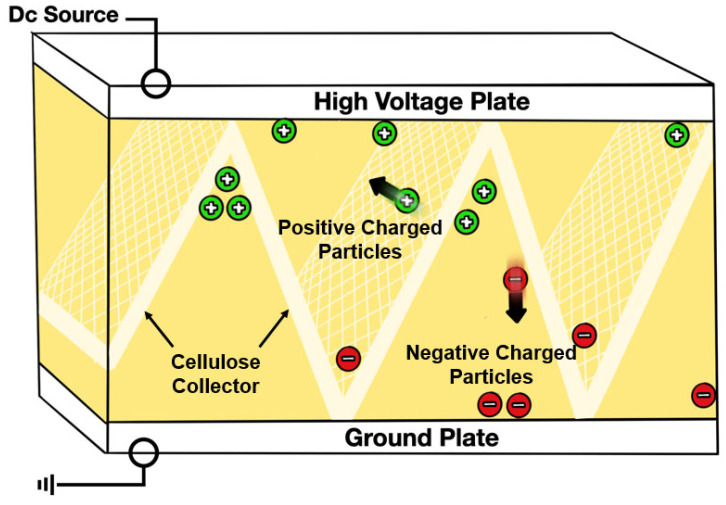
Electrostatic oil cleaning.

**Figure 13 materials-16-03737-f013:**
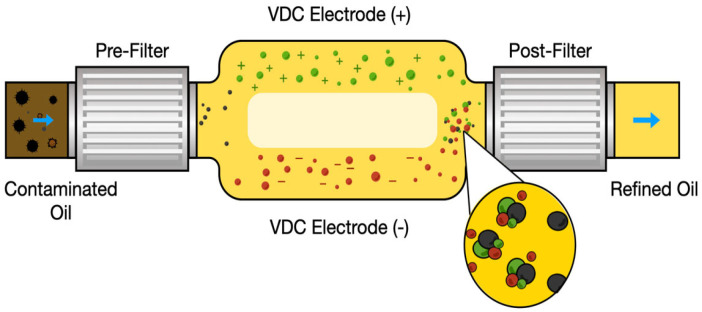
Charged agglomeration cleaning.

**Table 1 materials-16-03737-t001:** Deposit tests [21].

	Panel Coker [63]	Hot Tube [64,65]	Sliding Ring [63]	Micro-Coking [66]	Quick Coker [63]	HHI Test [67]
Test temp.	310–330 °C	300 °C	300–350 °C	330 °C	330 ± 10 °C	220 ± 5 °C
Oil temp.	100 °C	-	125–130 °C	-	-	-
Oil film forming	Splash	Cycling in hot tub	Splash	Temperature gradient	Drop	Pour
Sample volume (mass)	250 mL	5 g	250 mL	0.05 mL	10 mL	10 mL
Oxygen flow	-	1.4 mL/min	-	-	-	-
Measurement parameters	Varnish merit	Deposit merit	Varnish merit	Time of deposit forming	Varnish merit	Deposit and pull-off pressure
Test time	6 h	4 h	2 h	Several hours	2.5–4 min	20 ± 5 min
Set-up cost	High	High	High	Low	Low	Low

**Table 2 materials-16-03737-t002:** Chemical cleaning and solvency improving products.

Technologies	Chemical Cleaning	Solvency Enhancer
Product (Manufacturer)	Mobil™ System Cleaner (ExxonMobil) [75]	BOOST VR+ (Fluitec) [76]
Kinematic Viscosity @40 °C	55 mm^2^/s	54 mm^2^/s
Density @15 °C	0.926 kg/m^3^	0.86 kg/m^3^
Appearance	Clear Brown	White
Flash Point	138 °C	233 °C

**Table 3 materials-16-03737-t003:** Comparison of products related to varnish removal technology.

Technology	Depth Filtration	Electrostatic Oil Cleaning	Charge Agglomeration Cleaning	Resin Based Chemical Absorption
Product (Manufacturer)	Varnish Removal Unit with Varnish Removal insert 27/27 Filter Insert (CJC) [80]	Green Macheen™ 300 (Oilkleen) [81]	SMR 10 (Parker) [82]	Pall Sentry™ (PALL) [83]
Oil Reservoir Volume, max.	45,000 L	38,000 L	-	-
Oil Viscosity	68 mm^2^/s	100 mm^2^/s	220 mm^2^/s	500 mm^2^/s
Flow Rate, max.	32 L/min	20.52 L/min	38 L/min	11.36 L/min
Varnish Holding Capacity	8 kg	9 kg	-	-
Max. Oil Temperature	105 °C	82 °C	93 °C	59.4 °C
Design Pressure	4 bar	-	3.4 bar	5.4 bar
Filtration Resolution	3 µm	0.01 µm	5 µm	12 µm

## Data Availability

Not applicable.

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
