# Peer review of "Varnish Formation and Removal in Lubrication Systems: A Review"

_materials, 2023, doi:10.3390/ma16103737_

Round 1
Reviewer 1 Report
In this review, the definitions and characteristics, generating machinery, generating mechanisms, causes, measurement methods, and prevention or removal methods of varnish are summarized. It’s valuable for understanding the problem about varnish and helpful to those who are interested in oil investigation.
But, in my opinion, this review is only a combination of several reports about varnish, like a textbook. Reasonable analysis for related literature is absent, and the introduction is not deep enough. For example, several removing-varnish methods should be compared, rather than different manufacturer products.
In my opinion, this topic could not be regarded as a literature review for varnish. Even though many references are mentioned, only several basic information about varnish is present, without enough analyzing into these references. If there is any chance for this manuscript, I think the author could change the topic, related to general information about varnish in lubrication system, rather than literature review. From this perspective, this manuscript is valuable and could be accepted by me.
Author Response
Dear Editor and Reviewer
We appreciate your comments.
We prepared sincere responses to your comments as follows.
In this review, the definitions and characteristics, generating machinery, generating mechanisms, causes, measurement methods, and prevention or removal methods of varnish are summarized. It’s valuable for understanding the problem about varnish and helpful to those who are interested in oil investigation.
But, in my opinion, this review is only a combination of several reports about varnish, like a textbook. Reasonable analysis for related literature is absent, and the introduction is not deep enough. For example, several removing-varnish methods should be compared, rather than different manufacturer products.
Answer: We have experience in developing devices that remove contaminants from lubricants such as varnish and moisture. In this process, we investigated previous literature about varnish. In the process, we couldn't find enough papers that summarized the results of previous research on varnish or related technologies. Moreover, it was thought that review paper about varnish was necessary.
I have also published a paper about a method for measuring varnish contamination based on a lubricant sensor. We plan to continue research on technology development for varnish removal and condition diagnosis methods for varnish contamination. I tried to prepare this review paper based on my research experience on varnish and my research experience on lubricants for more than 10 years. However, most of the contents about varnish were published in magazines or reports, so it was difficult to organize them. Of course, there may be some shortcomings in this paper that we have organized. This research was conducted in the hope that this paper would be helpful to those who do related work or research in the future.
In consideration of the reviewer’s comments, we have revised and supplemented as follows.
[Introduction]
Furthermore, it is intended to present a new direction for research on varnish, such as evaluating varnish contamination and solving problems caused by varnish at an early state by using condition diagnosis technology based on oil sensors, which is widely applied these days.
[2.4 Causes]
There are five classification of base stock in API (American Petroleum Institute). Group 1 is the least refined and usually a mix of different hydrocarbon chains. These oils are generally used in applications without high performance demands. Group I oils are more polar in nature. This polarity leads to higher solvency than other mineral oil base stocks. Group II and Group III oils are more refined than Group I, and which are more oxidatively stable. However, the additives needed to increase solvency deplete, the other additives begin to separate and form varnish. That is, the group of base oil also affects the formation of varnish [39].
[2.6. Preventing or removing methods]
Although there are various techniques for removing varnish as described above, however, the problems caused by varnish contamination have not yet been well resolved. One of the reasons is that regular off-line analysis method such as MPC are not effective to diagnose varnish contamination immediately. Therefore, it is required to develop a method for accurately measuring varnish contamination in real time using as lubricant sensor. Moreover, it is also necessary to reduce maintenance costs by improving the durability and performance of the filtering systems.
[3. Conclusions]
Basically, it is necessary to improve the performance of filtering systems. Moreover, a device that effectively remove varnish in a melted stat as well as a technology for diagnosing varnish contamination at an early stage using lubricant sensors should be combined.
We did English proofreading once more by a professional company.
We revised manuscript in consideration of the reviewer’s comments. We would appreciate it if you could review the revised manuscript once more.
It is thought that the completeness of our paper has been improved more through the correction of each comment.
Thank you once again for reviewing our paper.
Sincerely yours,
Sung-Ho Hong

Reviewer 2 Report
This is a comprehensive literature survey of varnish in lubrication systems. It covers interesting and useful information on varnish generation mechanisms and causes, commonly affected machine components and measurement methods. The information on how to prevent and/or remove varnish is useful in practical applications. This survey is well structured and presented. It is very easy to follow. Since I have not seen such a summary paper on this topic, I find this work is useful. Having said that, the title of the manuscript may be changed to a literature survey, which is probably more appropriate to reflect the content of the current version. To be a good literature review paper, considerable amount of work on the most important present and future directions of varnish in lubrication systems needs to be included.
1. What is the main question addressed by the research? Varnish in lubrication system, including varnish generation mechanisms and causes, commonly affected machine components and measurement methods
2. Do you consider the topic original or relevant in the field? Does it address a specific gap in the field? It is a literature survey. I have not found a similar publication on this topic.
3. What does it add to the subject area compared with other published material? I think the work contains useful information on the topic, in particular, how to measure and prevent varnish.
4. What specific improvements should the authors consider regarding the methodology? What further controls should be considered? It would be good to add information on any knowledge gaps on the topic and possible approaches (with some details) to solve varnish-related issues.
5. Are the conclusions consistent with the evidence and arguments presented and do they address the main question posed? Yes, I think so.
6. Are the references appropriate? Yes
Author Response
Dear Editor and Reviewer
We appreciate your comments.
We prepared sincere responses to your comments as follows.
This is a comprehensive literature survey of varnish in lubrication systems. It covers interesting and useful information on varnish generation mechanisms and causes, commonly affected machine components and measurement methods. The information on how to prevent and/or remove varnish is useful in practical applications. This survey is well structured and presented. It is very easy to follow. Since I have not seen such a summary paper on this topic, I find this work is useful.
Having said that, the title of the manuscript may be changed to a literature survey, which is probably more appropriate to reflect the content of the current version. To be a good literature review paper, considerable amount of work on the most important present and future directions of varnish in lubrication systems needs to be included.
Answer: We improved the title as follows.
Varnish Formation and Removal in Lubrication Systems: A review
In addition, we added contents as follows:
(Introduction part) Furthermore, it is intended to present a new direction for research on varnish, such as evaluating varnish contamination and solving problems caused by varnish at an early state by using condition diagnosis technology based on oil sensors, which is widely applied these days.
(Conclusion part) Basically, it is necessary to improve the performance of filtering systems. Moreover, a device that effectively remove varnish in a melted stat as well as a technology for diagnosing varnish contamination at an early stage using lubricant sensors should be combined.
Point-1: What is the main question addressed by the research? Varnish in lubrication system, including varnish generation mechanisms and causes, commonly affected machine components and measurement methods
Point-2: Do you consider the topic original or relevant in the field? Does it address a specific gap in the field? It is a literature survey. I have not found a similar publication on this topic.
Point-3: What does it add to the subject area compared with other published material? I think the work contains useful information on the topic, in particular, how to measure and prevent varnish.
Point-4: What specific improvements should the authors consider regarding the methodology? What further controls should be considered? It would be good to add information on any knowledge gaps on the topic and possible approaches (with some details) to solve varnish-related issues.
Answer: We added contents as follows:
(Main subject : Varnish part)
Although there are various techniques for removing varnish as described above, however, the problems caused by varnish contamination have not yet been well resolved. One of the reasons is that regular off-line analysis method such as MPC are not effective to diagnose varnish contamination immediately. Therefore, it is required to develop a method for accurately measuring varnish contamination in real time using as lubricant sensor. Moreover, it is also necessary to reduce maintenance costs by improving the durability and performance of the filtering systems.
Point-5: Are the conclusions consistent with the evidence and arguments presented and do they address the main question posed? Yes, I think so.
Point-6: Are the references appropriate? Yes
We appreciate your comments during the first paper review process.
It is thought that the completeness of our paper has been improved more through the correction of each comment.
Thank you once again for reviewing our paper.
Sincerely yours,
Sung-Ho Hong

Reviewer 3 Report
Original Submission
Recommendation: Minor Revision
Comments to Author:
Manuscript ID: materials-2291836
Type of manuscript: Review
Title: A Literature Review of Varnish in Lubrication System
In this review, authors present an investigation of varnish contamination in lubrication systems. Definition, mechanisms, causes and methods of measuring are provided as well as preventing or removing methods of the problems associated at the presence of varnish in industrial oil systems. Because of this, the current study is a topic of relevance and general interest to the readers of Materials section: Thin Films and Interfaces.
The work is written in good technical language and a wide investigation about varnish-related problems has been conducted, however, some aspects should be improved to publish this paper:
- References in the text are not in the appropriate place, principally, in section 2.5. For example, in lines 379 and 380, the experimental procedure used for ultracentrifuge oil is mentioned, however, it is not possible to know who employed that method between the four references listed at the end of the paragraph. This problem is found also in RULER method, DSC…
- Size of Figure 2 should be enlarged.
- Please, review English spelling, mainly hyphenation, e.g., lines 169,170,172, 173…
- It would be necessary to explain differences between Group I and Group II oils in the text.
- Figure 6 must include all methods of measuring varnish potential mentioned in section 2.5: UC, CPA…
- Please, improve quality and definition of Figure 9.
Author Response
Dear Editor and Reviewer
We appreciate your comments.
We prepared sincere responses to your comments as follows.
In this review, authors present an investigation of varnish contamination in lubrication systems. Definition, mechanisms, causes and methods of measuring are provided as well as preventing or removing methods of the problems associated at the presence of varnish in industrial oil systems. Because of this, the current study is a topic of relevance and general interest to the readers of Materials section: Thin Films and Interfaces.
The work is written in good technical language and a wide investigation about varnish-related problems has been conducted, however, some aspects should be improved to publish this paper:
Point-1: References in the text are not in the appropriate place, principally, in section 2.5. For example, in lines 379 and 380, the experimental procedure used for ultracentrifuge oil is mentioned, however, it is not possible to know who employed that method between the four references listed at the end of the paragraph. This problem is found also in RULER method, DSC…
Answer: We revised reference list and arrange.
Point-2. Size of Figure 2 should be enlarged.
Answer: We enlarged the size of Figure 2.
Point-3: Please, review English spelling, mainly hyphenation, e.g., lines 169,170,172, 173…
Answer: We corrected wrong spell and hyphenation.
Point-4: It would be necessary to explain differences between Group I and Group II oils in the text.
Answer: We added explanation about differences between Group I and Group II oils as follows:
There are five classification of base stock in API (American Petroleum Institute). Group 1 is the least refined and usually a mix of different hydrocarbon chains. These oils are generally used in applications without high performance demands. Group I oils are more polar in nature. This polarity leads to higher solvency than other mineral oil base stocks. Group II and Group III oils are more refined than Group I, and which are more oxidatively stable. However, the additives needed to increase solvency deplete, the other additives begin to separate and form varnish. That is, the group of base oil also affects the formation of varnish [39].
Point-5: Figure 6 must include all methods of measuring varnish potential mentioned in section 2.5: UC, CPA…
Answer: We revised Figure 6 including all methods.
Point-6: Please, improve quality and definition of Figure 9.
Answer: We revised definition of Figure 9 as follows:
Figure 9. Reference of ultracentrifuge (UC) test [62].
But we can’t improve quality of the figure because we captured the figure in website.
We appreciate your comments during the first paper review process.
It is thought that the completeness of our paper has been improved more through the correction of each comment.
Thank you once again for reviewing our paper.
Sincerely yours,
Sung-Ho Hong
